# Penetrating Cardiac Injuries: Outcome of Treatment from a Level 1 Trauma Centre in South Africa

**Shumani Makhadi** [1,*] **, Maeyane Stephens Moeng** [1,2] **, Chikwendu Ede** [1] **, Farhana Jassat** [1]
**and Sechaba Thabo Palweni** [1]

1   Department of Surgery, Trauma Unit, Charlotte Maxeke Academic Hospital, University of Witwatersrand, Johannesburg 2193, South Africa; drmoeng.trauma@gmail.com (M.S.M.); chikwendu.ede1@wits.ac.za (C.E.); farhana.jassat1@wits.ac.za (F.J.); stpalweni@yahoo.com (S.T.P.)
2   Milpark Hospital, Johannesburg 2193, South Africa
*   Correspondence: drmakhadis@gmail.com

**Abstract:** Background: Penetrating cardiac injuries are rare in South African and international literature. Penetrating cardiac injuries are regarded as one of the most lethal injuries in trauma patients. The mechanism of injury varies across the world. In developing countries, stab wounds cause the majority of penetrating cardiac injuries. These injuries remain clinically challenging and are associated with high mortalities. Aim: To describe our experience with penetrating cardiac injuries and the outcome of their management at a level 1 trauma unit in Johannesburg, South Africa. Materials and methods: We retrospectively reviewed all patients who presented with penetrating cardiac injuries over a period of four years (1 January 2016 to 31 December 2019). The patients were identified using the hospital database. The patient's demographics, mechanism of injury, injury severity score, vital signs, investigation findings, final diagnosis, type of operation, length of hospital stay, morbidities, and mortalities were recorded. Results: There was a total of 167 patients with penetrating cardiac injuries identified. There were 151 (90.4%) males, with an overall median age of 29 years (IQR 24–34). Stab wounds accounted for 77.8% of the injuries, while gunshot wounds (GSW) accounted for 22.2%. The median injury severity score (ISS) and revised trauma score (RTS) were 25 and 7.1, respectively. The right ventricle was the most injured chamber (34.7%), followed by the left ventricle (29.3%), right auricle (13.2%), right atrium (10.2%), and combined injuries accounted for 7% of injuries. A commonly used incision was a sternotomy (51.5%), left anterior-lateral thoracotomy (26.9%), emergency room thoracotomy (19.2%), and clamshell thoracotomy (2.4%). The overall mortality rate was 40.7%, with a 29.2% mortality in the stab wounds. Twenty-four (14.4%) patients died in the emergency department, sixteen (9.6%) patients died on the table in theatre, and the remaining twenty-eight (16.7%) died in the intensive care unit or wards. Gunshot wounds, other associated injuries, right ventricle injuries, a high ISS, low RTS, and low Glasgow coma scale were all significantly more likely to result in death ($p < 0.001$). Conclusions: Penetrating cardiac injuries are often fatal, but the mortality can be improved with appropriate resuscitation and a work-up. The injuries to the heart can be safely managed by trauma/general surgeons in our setting. The physiology in presentation and other associated injuries determines outcomes in patients with penetrating cardiac injury.

**Keywords:** cardiac injury; stab wounds; gunshot wounds; thoracotomy

## 1. Background

Penetrating cardiac injuries are rare in South African and international literature [1,2]. In the United States, cardiac injuries accounted for 6.4% of all chest trauma in a study performed over 24 years [1]. Penetrating cardiac injuries are regarded as one of the most lethal injuries in trauma patients [1,2]. They remain clinically challenging injuries, though not unusual in our setting [1–4]. Thirty-nine patients, on average, presented annually with

penetrating cardiac injuries in a study previously conducted in Johannesburg [1,2]. It was previously reported that only 6–10% of patients presented to a hospital [2,3]. The mortality among these patients in-hospital has been described to be as high as 86% [4–7].

The distribution of mechanisms of penetrating cardiac injuries varies across the world. In South Africa, stab wounds are the leading cause of penetrating cardiac injuries. Campbell et al. reported 94% stab wounds and 6% gunshot wounds (GSW) in their series of penetrating cardiac injuries [3]. Degiannis reported 82% stab wounds and 18% GSW in Johannesburg [2]. However, in the United States, studies showed a 65% to 70% GSW and 30% to 35% of stab wounds [6,7].

Penetrating cardiac injuries affect more male patients than female ones in the literature [1–7]. The average age of these patients is 30 years old [2,5–9]. South African studies have shown mostly Africans being injured [2–4]. The clinical presentation of patients with penetrating cardiac injuries varies, and different presentations have been described [2,10].

Saadia et al. described five categories of clinical presentation: lifeless, critically unstable, cardiac tamponade, thoracoabdominal injury, and benign presentation [10]. This classification was developed to aid in the rapid diagnosis and swift management of penetrating cardiac injuries [10]. The revised trauma score (RTS) and injury severity score (ISS) have also been described in the approach and classification of penetrating cardiac injuries [6,7,11–13]. Other studies used clinical descriptions based predominantly on the haemodynamic parameters of the patients [8,9].

The time interval until management contributes towards outcomes in patients with penetrating cardiac injuries [14]. Most patients with penetrating cardiac injuries have been reported to die before arrival to a hospital. However, improvements in prehospital management and trauma systems are seeing more patients presenting to the emergency room [14–18]. These improvements included less time spent on scene and proper diversion to the appropriate level of trauma care a patient with penetrating cardiac injuries would need [14–18]. In a study conducted in Kwazulu-Natal, South Africa, in 1997, only 6% of the 1198 patients who were reviewed presented to the hospital with penetrating cardiac injuries. This number increased to 27% (76/282) in patients presenting to hospitals in 2011 [19].

There are acceptable indications for emergency room thoracotomy in the literature [20]. The rate of ERT varies widely in the world, from 9 to 70% [20–22]. ERT is the procedure of choice in a patient that arrests or remains hypotensive in the emergency department at our institution. Our previously reported survival rate for ERT with penetrating cardiac injuries was 40.6% [21]. The survival from ERT was better when performed for a tamponade [21]. The survival decreased in those who presented with high doses of adrenalin [21,22].

We have seen an increase in the number of penetrating cardiac injuries presenting to our unit in recent years [1–4]. Reasons for seeing an increase in penetrating cardiac injuries can be related to increased interpersonal violence or intent from the assailant [1–6]. The role of poverty, the widened gap between the rich and the poor, and the over-whelmed security cluster may be creating an environment that allows for the increased occurrence of penetrating cardiac injuries [23,24].

In our institution, trauma and general surgeons primarily manage penetrating cardiac injuries instead of cardiothoracic surgeons, as seen in other centres. On arrival in the emergency department, the patients are assessed following ATLS® guidelines by the trauma team led by a trauma surgeon. We hypothesize that penetrating cardiac injuries can have same outcomes even in resource-limited environments. Our study aims to describe our experience with penetrating cardiac injuries and outcomes of management at a level 1 trauma centre in Johannesburg, South Africa.

## 2. Materials and Methods

We, retrospectively, reviewed all patients who presented to Charlotte Maxeke Academic Hospital trauma unit with penetrating cardiac injuries over four years (1 January 2016 to 31 December 2019). The patients were identified from the hospital trauma database.

The patients' demographics, mechanism of injury, injury severity score (ISS), revised trauma score (RTS) vital signs on presentation, investigation findings, final anatomical site of diagnosis, type of operation, choice of surgical incision, length of hospital stay, morbidities, and in-hospital mortalities were recorded.

*Statistical Analysis*

Means ($\pm$SD) and median (interquartile range) were presented for continuous variables, and frequencies (%) were presented for categorical variables. All analyses were performed using STATA version 15. Continuous variables were first tested for normality using the Shapiro–Wilk test. Fisher's exact test was used to test for the significance of the relationship between categorical variables. Chi-square test was also used to test the relationship between categorical valuables. A multiple logistic regression analysis was performed to estimate independent predictive factors of death. *p*-value of <0.05 was considered statistically significant. The univariate association tests used were the Mann–Whitney U test (continuous variables) and Pearson Chi-square/Fisher's (categorical variables) to determine if significant differences existed between the two groups of the outcome variable (mortality).

Multivariate analysis was performed through stepwise forward and backward regression, modelled against the probability that the patient died (including odds ratios and 95% CI.

Ethics approval was obtained from the University of the Witwatersrand Human Ethics committee and the hospital' CEO. Ethics number M180463 was allocated to the study.

## 3. Results

There was a total of 167 patients with penetrating cardiac injuries identified. There were 90.4% males, with an overall median age of 29 years (range of 24 to 34 years). The mechanism of injury was due to stab wounds in 77.8% of the cases versus 22.2% from GSW (Table 1).

**Table 1.** Demographics and physiological parameters.

| Variable | Total |
|:---:|:---:|
| | n = 167 (100%) |
| Age (years) | 29 (IQR 24–34) |
| Gender | Male = 151 (90.4%)<br>Female = 16 (9.6%) |
| Mechanism<br>Stab wound<br>Gunshot wound | <br>130 (77.8%)<br>37 (22.2%) |
| Prehospital vitals<br>Systolic BP (mmHg)<br>Diastolic BP (mmHg)<br>Pulse rate (beats/minute) | <br>98 (IQR 81–110)<br>63.8 $\pm$ 20.2<br>90 (IQR 70–103) |
| Emergency department vitals<br>Systolic BP (mmHg)<br>Diastolic BP (mmHg)<br>Pulse rate (beats/minute) | <br>90 (IQR 53–110)<br>59.7 $\pm$ 20.9<br>91 (IQR 76–112) |
| pH<br>Lactate (mmol/L)<br>Base excess (mmol/L)<br>ISS<br>RTS | 7.2 $\pm$ 0.15<br>6.6 (IQR 4–10)<br>$-8.75$<br>25 (IQR 25–29)<br>7.1 (IQR 4.5–7.1) |

The prehospital vital signs showed a median systolic blood pressure of 98 mmHg, a mean diastolic blood pressure of 63 mmHg, and a median heart rate of 90 beats per

minute (bpm) (Table 1). The median ISS was 25, with a median RTS of 7.1. On presentation to the emergency department, the arterial blood gas showed an average acidaemia of 7.2, with a lactate and base excess of 6.6 and −8.8 mmol/L, respectively (Table 1). The median blood pressure in the emergency department was 90 mmHg, the mean diastolic pressure was 59 mmHg, and the heart rate was 91 bpm (Table 1). Beck's triad was absent in 52% of patients. Sixty-four percent of patients presented haemodynamically/metabolically unstable. Fifty-nine patients presented with a systolic blood pressure of less than 90 mmHg.

The diagnosis was determined clinically in some of the patients with obvious isolated stab wounds to the chest, others were identified with the use of an e-FAST. The investigations performed in the emergency department were a chest X-ray (CXR) (64%), extended focused assessment with sonography for trauma (eFAST) (29%), central venous pressure insertion (CVP) (33%), electrocardiography (ECG) (8%), cardiac enzymes (16%), and emergency echocardiogram (25%). Sixty-four percent of patients had associated chest injuries, twenty-two percent had abdominal injuries, and ten percent had associated head injuries (Table 2). The most common organ injured in the chest was the lung. The liver was the most common organ injured in the abdomen.

**Table 2.** Additional associated injury sites.

| Site | Number of Patients (Percentage) |
|---|---|
| Chest | 108 (64.7%) |
| Haemothorax | 20 (12%) |
| Pneumothorax | 10 (6%) |
| Parenchymal injury | 68 (40.7%) |
| Internal mammary | 10 (6%) |
| Abdomen | 37 (22.2%) |
| Solid organ | 12 (7.2%) |
| Hollow viscus | 11 (6.6%) |
| Solid and hollow viscus | 14 (8.4%) |
| Extremities | 21 (12.6%) |
| Head injury | 17 (10.2%) |
| Minor | 7 (4.2%) |
| Major | 10 (6%) |

The right ventricle was the most injured chamber (34.7%), followed by the left ventricle (29.3%), right auricle (13.2%), right atrium (10.2%), left auricle (1.8%), and left atrium (1.2%), and 7% of patients had combined injuries (Table 3). Four patients sustained injuries to the coronary artery (Table 3). Most patients had a sternotomy (51.5%), left thoracotomy (26.9%), emergency room thoracotomy (19.2%), and clamshell thoracotomy (2.4%) (Table 4). Prolene 2/0 was the preferred suture for the ventricles, whereas prolene 4/0 was used to repair the auricles and atrium. Pledgets were only used in atrial and auricular injury repairs (n = 16). All patients who survived to the ICU had an echocardiogram in-hospital to exclude valvular, papillary, and interventricular injuries before discharge.

**Table 3.** Site of cardiac injury.

| Site of Injury | Number of Patients (Percentage) N = 167 |
|---|---|
| Right ventricle | 58 (34.7%) |
| Left ventricle | 49 (29.3%) |
| Right auricle | 22 (13.2%) |
| Right atrium | 17 (10.2%) |
| Combined injuries | 12 (7.2%) |
| Left auricle | 3 (1.8%) |
| Left atrium | 2 (1.2%) |
| Coronary vessel injuries | 4 (2.4%) |

**Table 4.** Type of incision.

| Incision | Patients (Percentage) |
|---|---|
| Sternotomy | 86 (51.5%) |
| Thoracotomy total | 81 (48.5%) |
| Emergency room thoracotomy | 32 (19.2%) |
| Left anterior lateral thoracotomy | 45 (26.9%) |
| Clamshell thoracotomy | 4 (2.4%) |

Complications were coagulopathy (14%), surgical site sepsis (1%), empyema (2%), and arrhythmia 0.61% (n = 1). The overall mortality rate was 40.7% (n = 68/167). Twenty-four (14.4%) patients died in the emergency department, sixteen (9.6%) patients died on the table in theatre, and the remaining twenty-eight (16.7%) patients died in the ICU or wards (Table 5). The mortality rate from gunshot wounds was 81.1% (n = 30/37) and 29.2% (n = 38/130) from stab wounds. All survivors were admitted to the ICU. Forty-one percent of patients were extubated in the operating room, and the rest extubated in ICU. Among the survivors, eighty-six percent (n = 85/99) of patients were discharged home, and fourteen percent (n = 14/99) of patients were stepped down to other facilities to continue care.

**Table 5.** Mortality outcome's variables.

| Variable | Survivors N = 99 | Nonsurvivors N = 68 | *p* Value |
|---|---|---|---|
| Mechanism | | | |
| Gunshot | 7 (7.1%) | 30 (44.1%) | |
| Stab | 92 (92.1%) | 38 (55.9%) | <0.001 |
| Associated injuries | | | |
| Chest | 60 (60.6%) | 43 (63.2%) | 0.001 |
| Abdomen | 11 (11.1%) | 23 (33.8%) | <0.001 |
| Extremities | 3 (3%) | 11 (16.2%) | 0.002 |
| Head injury | 2 (2%) | 12 (17.6%) | <0.001 |
| Systolic BP < 90 mmHgΔ (casualty) | 29 (29.3%) | 30 (44.1%) | 0.002 |
| Systolic BP > 90 mmHgΔ (casualty) | 46 (46.5%) | 14 (20.6%) | 0.002 |
| No BP measurable (casualty) | 24 (24.2%) | 24 (35.3%) | |
| pH | 7.23 ± 0.12 | 7.13 ± 0.18 | 0.002 |
| Lactate | 6.00 (3.4–8.6) | 7.45 (5.4–13.7) | 0.027 |
| Base excess | −7.8 (−11.9, −4.3) | −10.7 (−18, −6.4) | 0.019 |
| ISS | 25 | 27 | <0.001 |
| RTS | 7.8 | 5.6 | <0.001 |
| GCS | 15 | 12 | <0.001 |
| Outcomes | Discharged home (85 (52.8%)) Transferred (14 (8.7%)) | Died in Casualty (24 (14.4%)) Died on the table (16 (9.6%)) Died in ICU/ward (28 (16.7%)) | |

Among the nonsurvivors, gunshot wounds, associated injuries, RV injuries, a high ISS, low RTS, hypotension in casualty, and low GCS were all negative predictors of mortality. (Table 5). Using a multivariate regression analysis, associated abdominal injuries were the variate with the highest risk of mortality (OR: 4, *p* < 0.001).

All patients were operated on by trauma surgeons/general surgeons or residents under the direct supervision of the trauma surgeon at our institution. None of the patients needed a cardiopulmonary bypass, which would have required a cardiothoracic surgeon.

## 4. Discussion

Most of our patients were young males, which was in keeping with most trauma literature [1–6]. We still experience a lot of interpersonal violence in South Africa compared to other developing countries [2,25]. We saw an increasing number of penetrating cardiac injuries despite efforts to try and prevent these [10]. A study conducted at the Chris Hani Baragwanath hospital in 2006 almost had the same amount of penetrating cardiac injuries as our study [2]. One would have expected these injuries to be on a downward trend. Injuries were sustained after stab wounds (77.8%) and GSW (22.2%). This was in keeping with the distribution of penetrating cardiac injuries in South Africa [2,3,15,19,25]. Knives are readily available, which was the most used weapon in our cohort.

On presentation to the emergency room, the presence of a Beck's triad (hypotension, tachycardia, and muffled heart sounds) in penetrating cardiac injuries had poor sensitivity; even the presence of one of the parameters in our patients had poor sensitivity [15,25]. We had 68% of patients present with more than one of the Becks triad parameters. This was lower than the number described in another facility in Johannesburg, which was 78% [1,9]. The most common findings were tachycardia and hypotension, which most trauma patients presented from other causes. One must have a high index of suspicion when evaluating patients with suspected penetrating cardiac injuries.

Most of our patients presented with hypotension and acidaemia. The patients' physiology shows the severity of the injuries and the stress a penetrating cardiac injury has on the body. Mina et al. demonstrated in previous reports that patients with a high ISS and low RTS were associated with poor outcomes [7,17]. Most of the patients were young males. Their ability to compensate on a haemodynamic level was remarkable. The blood pressures on presentations were lower than the prehospital ones, which indicated that patients deteriorated en-route to our trauma unit. Hypotension (systolic BP < 90 mmHg) was associated with mortality ($p = 0.002$), which was derived from blood pressure obtained in a hundred and nineteen patients in casualty. The time to hospital presentation was associated with better outcomes. The deterioration might be explained by resource-limited prehospital services [26].

The identification of patients with penetrating cardiac injuries can be challenging at times [2–6,15]. Chest X-rays, central venous pressure, extended FAST, cardiac enzymes, ECG, and echocardiogram have all been described to identify injuries to the heart [1–6]. A pericardial window remains the gold standard for diagnosing a potential cardiac injury after penetrating chest trauma [1,6,15]; however, it is invasive and requires general anaesthesia. The emergence of an extended FAST for the evaluation of pericardial fluid has good sensitivity and a positive predictive value [26,27]. We defined a positive extended FAST as a haemopericardium of more than 5 mm in size. Other causes of pericardial effusion should always be borne in mind in our setting, due to the associated HIV prevalence and possible TB pericarditis.

It is also essential to pick occult cardiac injuries; diagnosing them can also be challenging [6,15]. Some of these patients can develop a haemorrhage from two days and up to three weeks postinjury [6]. We had two patients who were of interest. We picked up their injuries based on their elevated Troponins trends. These patients were admitted to the ward for further investigations.

An emergency room thoracotomy was performed in 32 (19.2%) patients, and 4 (2.4%) patients had a clamshell thoracotomy. The most common reason for ERT was a witnessed arrest in the emergency department and systolic BP < 60 mmHg despite resuscitation. Theatre is a considerable distance from our emergency room, so one must be careful with rushing patients to theatre as they may arrest en-route. A previous study in our institution reported an ERT survival rate of 40.6% in cardiac injuries [21]. Of the patients that went to theatre, the majority had a sternotomy, followed by a thoracotomy. The decision was left to the surgeon as to which incision to use. Thoracotomy was chosen most times to address combined heart–lung injuries. In South Africa, sternotomy is a commonly used incision, whereas, in the United States, a thoracotomy was favoured for penetrating

cardiac injuries [2,7–9]. About 20% needed a laparotomy to address associated abdominal injuries. Abdominal injuries in combination with a cardiac injury were associated with poor outcomes in our cohort (see Table 5).

The RV was the most injured chamber of the heart. This injury pattern was in keeping with other reports in the literature [1–4]. The reason for the right side being commonly injured might be related to the fact that most assailants are right-handed and that the right ventricle covers the greatest surface of the anterior chest wall [1–7,14]. The LV, right auricle, RA, left auricle, and LA were injured in 29.3%, 13.2%, 10.2%, 1.8%, and 1.2% of patients, respectively. This distribution was in keeping with previously reported studies [2,15]. Combined injuries were not common, making up less than eight percent of total injuries reported. However, one must always look posteriorly when evaluating an injured heart to avoid missing posterior surface injuries. Coronary artery injuries are often fatal without access to an emergency cardiopulmonary bypass [28]. We had four patients who sustained coronary artery injuries; three survived as their injuries were to the distal coronary artery. A cardiothoracic backup for an emergency bypass is available at our institution. In the absence of an emergency cardiopulmonary bypass, most proximal coronary artery injuries would have a high mortality rate [28].

Patients who sustained penetrating chest injuries also sustained other injuries in the chest apart from cardiac injuries. The haemothorax was the most commonly reported chest injury in 12%, and only 6% of patients sustained a pneumothorax injury (Table 2). Intraoperatively, the most commonly injured viscera were the lungs (40.7%) and the internal mammary artery (6%). A study by Clarke et al. also showed associated injuries to the chest with penetrating cardiac injuries, especially to the haemothorax [19]. A subset of patients presented with a negative e-FAST and a haemothorax injury. This false-negative result usually happened when the pericardium decompressed into the chest cavity (n = 2). The evaluation of patients with penetrating chest injuries involving the haemothorax and pneumothorax must also highlight and prompt the possibility of a cardiac injury [27].

Approximately 33 patients needed a laparotomy in addition to a sternotomy/thoracotomy. The solid organs were the most injured (liver, spleen, and kidneys). In a previous study performed at the Chris Hani Baragwanath hospital, patients with thoracoabdominal injuries could sustain injuries to both the chest and abdomen [2,10]. The hollow viscus was also injured in 6% of the patients. The evaluation of a patient with a thoracoabdominal injury can also be challenging. The decision of which cavity to start with can also be a daunting task, as shown by Degiannis et al., since it is possible to sustain injuries to both cavities [2].

The morbidity rate was 14%. Fourteen patients (8.5%) became coagulopathic, requiring massive transfusion activation; three patients (1.8%) developed empyema and surgical site infections; one patient (0.6%) developed arrhythmia. This infection rate was low despite the emergency nature of the surgical intervention. We used an echocardiogram to exclude valve injuries and septal wall injuries before hospital discharge.

The patients who performed well intraoperatively were extubated in theatre (42%). This quick response indicates that with adequate resuscitation, the patient's ability to recover is remarkable. Of the patients that were not extubated, the majority were extubated the following day in the ICU once their physiology had improved.

Gender-based violence has been on the increase in South Africa [29,30]. This has been reported as an epidemic by the President of South Africa. We had 16 females with penetrating cardiac injuries in total, with their median age being 31 years (28–40 years). Females composed 10% of our cohort, and some were injured by their partners [31]. They had a shorter average hospital stay (4–5 days) than their male counterparts, having mostly sustained lower-grade injuries. Their median ISS was 25 (range 25–31). The mortality rate was 2% in females.

The mortality for patients with stab wounds was lower compared to gunshot wounds. The overall mortality rate from stab wounds was in keeping with other studies performed in a setting such as ours [2]. The lower mortality rates from stab wounds were attributed to applying protocols for penetrating cardiac injuries even with junior staff present [2]. The

presence of a senior trauma consultant on the premises also made it possible to manage these patients urgently and appropriately [2].

GSW, a high ISS, low RTS, and acidaemia on presentation were statistically associated with mortality (see Table 5). Base excess, lactate, casualty diastolic BP, and gunshots showed significance association with mortality (Table 6). A single prehospital incident of hypotension was not associated with an increase in mortality. The association with head and abdominal injuries in patients with cardiac injuries was also associated with nonsurvivability (see Table 5). Patient physiology and associated injuries were good predictors of mortality in penetrating cardiac injuries [11–14].

**Table 6.** Logistical regression for variables that predicted mortality.

| Variable | Odds Ratio | *p*-Value | 95% Confidence Interval |
|---|---|---|---|
| Base excess | 1.22 | 0.009 | 1.05–1.42 |
| Lactate | 1.56 | 0.025 | 1.06–2.30 |
| Casualty diastolic BP | 0.94 | 0.033 | 0.89–1.00 |
| MOI | 0.02 | 0.001 | 0.002–0.20 |

## 5. Limitations of the Study

This was a retrospective study looking at a database with associated shortcomings. Some of the shortcomings were, this is just an audit of a single trauma unit, and no real conclusions could be drawn. The findings of this study could have been influenced by selection bias, as only patients reaching the hospital were studied, while autopsy data were not included in this study. There was no long-term follow-up beyond the initial standard outpatient review within three weeks of discharge.

## 6. Conclusions

Penetrating cardiac injuries are often fatal, but the mortality can be improved with appropriate resuscitation and a work-up. The injuries to the heart could be safely managed by trauma/general surgeons in our setting. The physiology on presentation and other associated injuries determined outcomes in patients with penetrating cardiac injuries.

**Author Contributions:** Conceptualization, M.S.M. and S.M.; methodology, F.J. and C.E.; data collection, S.T.P.; writing, M.S.M. and S.M. All authors have read and agreed to the published version of the manuscript.

**Funding:** This research received no external funding.

**Institutional Review Board Statement:** The study was conducted in accordance with the Declaration of Helsinki. Ethics approval was obtained from the University of the Witwatersrand Human Ethics committee and the hospital' CEO. Ethics number M180463 was allocated to the study.

**Informed Consent Statement:** Patient consent was waived due to the retrospective nature of the study.

**Data Availability Statement:** Data can be made available upon request from the authors.

**Conflicts of Interest:** The authors declare no conflict of interest.

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
