# Peer review of "Penetrating Cardiac Injuries: Outcome of Treatment from a Level 1 Trauma Centre in South Africa"

_traumacare, doi:10.3390/traumacare2020021_

Round 1

Reviewer 1 Report

This retrospective review article aims to describe the outcomes of penetrating cardiac trauma and management outcomes at a single level 1 trauma center in South Africa over 4 years. The conclusions that were drawn are the following: outcomes can be determined by physiology and associated injuries on presentation, mortality can be improved with appropriate resuscitation and workup. It is also seen that there is a high rate of ED thoracotomies being performed at this institution.

Strengths: This article provides insight to the South African trauma system where there is a larger number of penetrating cardiac injury compared to other nations. It illustrates their allocation of resources, workup and management of these patients who generally have high morbidity and mortality.

Weaknesses: Single center review. Only patients who survived to present to hospital were included, biasing results.

Overall: Interesting paper that shows the breadth of the situation of violent trauma in South Africa, and its management. Also makes comparisons with the American experience of penetrating trauma, which provides insights as to the differences. Aside from the minor corrections for figures and missing tables, it is well written.

Comments:

  1. P3L119, perhaps authors meant 22.2% for GSW
  2. 2. P4L135-137, the quoted percentages do not correspond to Table 2 figures.
  3. 3. P5L141, right ventricle written as 34.2% but on table 3 it is 34.7%
  4. 4. P6L163, where is table 6?
  5. 5. P7L181-183, unclear sentence, even the presence of one of the parameters in our patients? Seems incomplete.
  6. 6. P8L226, did you mean table 5? Please use a common numbering system either Roman or Arabic numerals throughout the paper.
  7. 7. P9L243, did you mean 12% sustained hemothorax, according to table 2?
  8. 8. P9L244, please reconcile these percentages with table 2.
  9. 9. P9L282 and L284, there is no table 6. Also, use a common numbering system throughout the paper.

Reviewer 2 Report

Thanks for the opportunity to review this manuscript on penetrating cardiac injuries.

Overall comment:

Although I recognize the effort that has been put into collecting the data and reporting it, what does this analysis add to what is already known? GSW are worse than stab wounds, the right ventricle is the most commonly injured part of the heart, sternotomy is the exposure of choice, cardio-pulmonary bypass is not required – none of this is new.

An audit of results can be useful for an institutional review of outcomes, however not particularly enlightening for clinical/scientific advancement. An institution’s “experience” does not lend itself to testing of a hypothesis. I could not see a hypothesis stated.

I would suggest that the structure of the manuscript could be made more succinct. I can see the attempt at describing what is known in the background, and also how the findings compare with existing literature in the discussion, however both sections are too long. There are also results presented in the discussion for the 1st time (these should be in Results). I have attached some recommended references for suggested manuscript formatting.

Justin Dimmick’s group

https://medicine.umich.edu/sites/default/files/content/downloads/WritingResearchPaper_Ibrahim_0.pdf

A table from Tim Pawlik’s group

Balch CM, McMasters KM, Klimberg VS, et al. Steps to Getting Your Manuscript Published in a High-Quality Medical Journal. Ann Surg Oncol. 2018;25(4):850-855. doi:10.1245/s10434-017-6320-6

TABLE 4

Common structure of clinical manuscripts

Title Write an informative title that will capture the reader’s interest. Titles should be a precise summary of the paper’s content, brief and clear. Avoid jargon and abbreviations

Abstract The abstract is the part that will be read first and most widely. Follow the journal format and word length for the abstract (typically about 250 words), and be sure to include the key data and conclusions

Key words (6–8) Key words help readers find your paper and support its listing to be at the top in search results. Avoid overlapping key words as well as key words that are only one word

Introduction The Introduction section should include 2 to 3 paragraphs about the background summarizing existing evidence and knowledge gaps, purpose of the study, and hypothesis (if the study was hypothesis driven) to convince readers that the study is significant and important

Methods Provide enough details (or citations if similar methods were reported in previous studies) so that others can judge the reliability of the study and reproduce your research. Clearly document the study design, study subjects, time period, metrics and outcomes, and statistical methods

Results Present the data and results germane to the hypothesis or the theme of the manuscript. Appropriate use of tables and figures is critical. Avoid repetition in the text of data presented in tables and/or figures. You may describe your objective interpretation of the data in the Results section rather than simply listing observations

Discussion Summarize the study and explain both the significance of the findings and how they fit in with previously published work. In other words, explain “what the findings mean.” We recommend writing a Discussion section that includes 5 paragraphs. In the first paragraph, briefly summarize the study results, their significance, and the originality of the findings. In the second paragraph, describe your study’s relationship to other studies. In the third paragraph, discuss specific findings of your choice (e.g., speculate about unexpected findings). In the fourth paragraph, describe the limitations of the study. In the final paragraph, briefly summarize the global conclusions, carefully avoiding overstatement of the study results. Although it is challenging, avoid excessive repetition of results between the Results section, Figures, and Discussion section

References List the publications cited in the manuscript. Use of citation software such as Endnote to insert and format the references is strongly recommended

Figure legends Write the legends such that they are brief and make the figures fully understandable without reference to the text of the manuscript

Specific comments:

How was the injury diagnosed ie what was the indication for surgery? Tamponade on e-FAST? What about the 70% that did not have an e-FAST. There is no mention of injury site – how many wounds were in the cardiac box? How many patients underwent a pericardial window?

Please be consistent with formatting – decimal points are written as commas in the manuscript; this should be corrected

“Among the non-survivors, gunshot wounds, associated injuries, RV injuries, high 162 ISS, low RTS, hypotension in casualty and low GCS were all negative predictors of mortality. (Table 6).” – there does not appear to be a table 6. Also if looking at predictors of mortality, was a multivariate analysis performed?

Round 2

Reviewer 1 Report

Corrections were made. No further comments, thank you.

Author Response

no response needed

Reviewer 2 Report

The general comments regarding hypothesis, justification for an audit, learning points and overall structure of the manuscript have not been addressed.

The discussion still has data that should be reported in the results section, and some results are repeated in the discussion. Again, please reconsider the structure of the manuscript and utilise the discussion to summarise and compare your findings to the literature ie how does it compare to what is already known?

I can't see where the question about diagnosis has been included in the manuscript.

I cannot see any inclusion of multivariate analysis in the manuscript - there are no odds ratios or confidence intervals, there is no description of adjustment

Author Response

see attachment for response

Round 3

Reviewer 2 Report

The description of diagnosis should not be in the discussion. It is part of the results and the specific numbers of how the diagnosis was made.

The following is an inadequate response:

"The statistician who did the analysis used the p value to show statistical significance in his analysis. We emailed him about the multivariate analysis and await his response."

Please perform and report the results of the multivariate analysis correctly

Author Response

The diagnosis was statement was moved to the results section. line 133 and 134

The multivariate analysis was added to the methods section. line 114 and 115

The multivariate results with the odds ratio added to the results. line 169 and 170

Round 4

Reviewer 2 Report

The reporting of the multivariate analysis is inadequate. No confidence intervals have been reported. No statement of adjustment for covariates has been made. I would suggest that resubmission be undertaken with an experienced biostatistician.

Author Response

after discussion with the statistician, please find the response
